# Road Rage as a Type of Violation of Well-Being in Traffic: The Case of Turkey

Zeynep Reva [1,*] and Oğuz Polat [2]

1  Department of Human Rights Law, Faculty of Law, Özyeğin University, 34794 İstanbul, Turkey
2  Department of Forensic Medicine, Faculty of Medicine, Acıbadem Mehmet Ali Aydınlar University, 34752 İstanbul, Turkey
*  Correspondence: zeynep.reva@ozyegin.edu.tr

**Abstract:** One of the essential components for understanding a life with dignity and with human rights is the right to health. The World Health Organization defines "health" as "a state of complete physical, mental and social well-being and not merely the absence of disease or infirmity". Physical and social well-being is not enough, and the individual's mental well-being should also be realized. Anger is one of the most important factors affecting the mental, as well as the physical, health of individuals. Anger can be both a cause and a consequence of poor mental health. Driving anger can be defined as the anger that occurs while driving, and its level can be associated with aggressive and risky driving, loss of concentration and vehicular control, and near miss accidents in traffic. In this research, the factorability of the 14-item short form of the Driver Anger Scale (DAS) was investigated in the Turkish population. The data were further analyzed for various demographics and independent variables. The short form of the DAS can be reliably used for Turkish drivers as well. A safe driving culture must be substantially popularized via educatory applications within digital or classroom environments to control, regulate, and lessen traffic violence. Effective audits and deterrent regulations are also important with respect to decreasing driving anger and violence.

**Keywords:** driver anger; violence in traffic; road rage; driver anger scale; driving anger expression

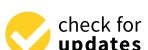



## 1. Introduction

The United Nations' Department of Economic and Social Affairs has declared sustainable development goals and criteria for both society and people. One of them is to ensure healthy lives and promote well-being for all ages. According to these sustainable targets, there are 17 main goals [1]. Goal 3.6 states "By 2020, halve the number of global deaths, and injuries from road traffic accidents". The main indicator for this goal is the death rate due to traffic injuries.

The right to health is a fundamental part of our human rights and of our understanding of a life lived in dignity. The World Health Organization (WHO) defines "health" as "a state of complete physical, mental and social well-being and not merely the absence of disease or infirmity" [2]. Physical and social well-being is not enough; the individual's mental well-being should also be realized. Anger is one of the most important factors affecting the mental, as well as the physical, health of individuals. Anger can be both a cause and a consequence of poor mental health.

Violence is a significant phenomenon of the 21st century that is often encountered in daily life. It threatens us in many dimensions [3]. Various examples of violence are encountered in traffic; from injuring to killing; from insults to threats; from approaching the bumper of another driver of the opposite gender to violating traffic safety. The most significant factor that leads to traffic violence, which can have serious consequences, is "road rage". It must be noted that there is no consensus on the definition of road rage. Having said that, road rage can be defined as "an incident where a driver or passenger

attempts to kill, injure, intimidate another driver or passenger, or damage their vehicle" [4]. It can also be defined as "a situation where a driver or passenger attempts to kill, injure, or intimidate a pedestrian or another driver, or cause damage to their vehicle in a traffic incident" [5] or "any driving behavior that deliberately endangers others psychologically, physically, or both" [6]. Driving anger is the main phenomenon that is encountered in road rage perpetrators.

There are no official data on traffic violence in Turkey. However, there are some data that are indirectly related to traffic violence. Road rage leads to unsafe behaviors that can end in accidents. Therefore, statistics on fatal accidents or accidents that result in serious injuries can provide an idea on the subject. According to the statistics from 2017 [7], there were 1,202,716 accidents in Turkey. Of them, 7427 people were killed and 300,383 people were injured. In Turkey, when certain newspaper reports and judicial decisions [8] (as per the Supreme Court records, 2013) were analyzed, it was found that, during traffic fights, tools including knives, guns, rifles, crowbars, baseball bats, and even the biting of ears [9] were used. Although baseball is not a common sport in Turkey and the rules of the game are largely unknown, Turkey is one of four countries in which the most baseball bats are sold [10].

Research shows that those with high driving anger levels are expected to become angry more frequently [11]. It has been specified that people who have a continuous anger temperament, as a personality trait, also have a high tendency to become angry while driving [12]. According to certain research, drivers who have a high degree of anger are more aggressive than others, behave in a risky manner, and may cause a fight. They do not use constructive manners of expression and do not think of consequences before responding to the other driver [11]. A Turkish study that was conducted on professional drivers stated that there were positive significant correlations between risky driving behavior and anger [13]. Another study from Turkey stated that disrespect from other drivers and driving at a slow speed also cause road rage [14]. Through this lens, traffic causes damage to our health. To avoid this situation, sustainable targets for healthy lives must be one of our main goals for daily life. Anger and stress are the main causes of unwanted results such as hypertension, headaches, heart diseases, neurological symptoms, and uncomfortable moods. Additionally, the stress and rage that traffic can cause has too many negative effects on the lives of people. The DAS is an approach for measuring driving anger. The original DAS questionnaire was developed in the USA by Deffenbacher et al. [11]. A cluster analysis of the 33-item scale produced six subscales (Figure 1).

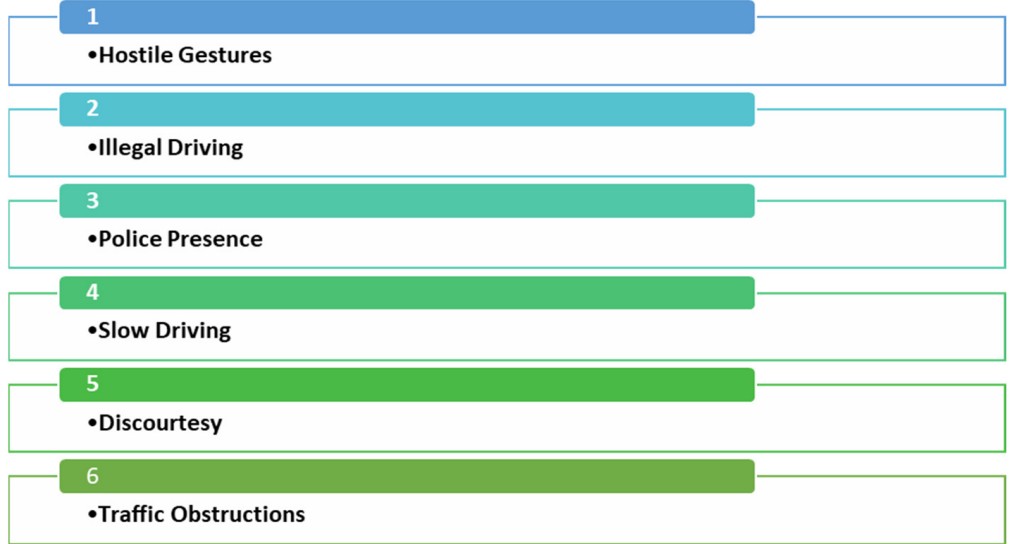

**Figure 1.** The six subscales of the 33-item scale.

A 14-item short form of the DAS was correlated with the DAS subscales. In the USA, by Deffenbacher et al. [11], four ways by which people can express their anger when driving were identified via a scale called the "Driving Anger Expression Inventory (DAX)". In this inventory are the following categories: "Verbal Aggressive Expression"; "Personal Physical Aggressive Expression"; "Use of the Vehicle to Express Anger"; and "Adaptive/Constructive Expression". Aggressive forms, which can be summarized into the Total Aggressive Expression Index, were noted to correlate positively with each other; however, they were uncorrelated or correlated negatively with adaptive/constructive expression. According to a study conducted in Turkey [14], a psychometric analysis of the scale showed that the Turkish-adapted DAX has the same factor structure; and that male drivers between 21 and 30 years old reported more physically aggressive expressions and used their vehicle to express more anger, whereas female drivers reported more adaptive/constructive expressions instead. In addition, based on the study in the USA [15] and the study in Malaysia [16], it was shown that young drivers are more likely to display anger and aggression. The research performed on the effects of driving experience on the DAS in New Zealand (NZ) [17] and in Malaysia [16] found that there was a negative correlation between drivers' driving experiences and the DAS. Based on the study by Deffenbacher [12], conducted with college students from the USA, it was shown that high anger drivers reported expressing their anger more through verbal, personal, physical, and vehicular aggressive expressions and less through adaptive/constructive means, than would be found in low anger drivers. In addition, males reported significantly more physical and vehicular aggressive expressions and fewer adaptive/constructive expressions than those reported by women. Men and women, however, did not differ on verbally aggressive expressions. The response to driving anger-eliciting situations can vary among cultures. Therefore, thus far, the DAS has been adapted into various cultures by researchers.

The aim of this study is to investigate and factor analyze the short form of the DAS and to adapt it for Turkish drivers. Furthermore, the data for this phenomenon were investigated for the relationships and differences between driver anger, driving anger expressions, demographics, and various other independent variables. The effects of an awareness of legal rights and responsibilities among drivers having low driving anger performance was also analyzed.

## 2. Materials and Methods

### 2.1. Participant Demographics and Various Independent Variables

In this study, there were 421 participants who were drivers in Turkey, including 220 females (52.3%) and 201 males (47.7%). The majority of them were from Istanbul (75.8%). They participated in an online survey voluntarily. Most of them were aged between 26–40 years old (44.4%), followed by 41–50 years old (33.0%). Further, the majority of respondents were university graduates (94.5%). More than half of the respondents had driving experience of 16+ years (51.1%). Furthermore, the majority of the participants drove their own personal cars (90.7%);their cars being either 3–10 years old (50.4%) or 0–3 years old (40.4%). In addition, they mostly drove weekdays between home and work (47.7%), and on every hour of the day (32.5%). Table 1 shows the participants' demographics and characteristics in detail.

**Table 1.** Participants' demographics and characteristics.

| Participants | Frequency | Percentage % | Cumulative Percent % |
|---|---|---|---|
| *Gender* | | | |
| Female | 220 | 52.3 | 52.3 |
| Male | 201 | 47.7 | 100.0 |
| *Age* | | | |
| 18–25 years | 12 | 2.9 | 2.9 |
| 26–40 years | 187 | 44.4 | 47.3 |
| 41–50 years | 139 | 33.0 | 80.3 |
| 51–60 years | 61 | 14.5 | 94.8 |
| 61+ years | 22 | 5.2 | 100.0 |
| *Education* | | | |
| <University | 23 | 5.5 | 5.5 |
| University | 398 | 94.5 | 100.0 |
| *Profession* | | | |
| Lawyer | 31 | 7.4 | 7.4 |
| Other | 136 | 92.6 | 100.0 |
| *City Lived* | | | |
| Istanbul | 319 | 75.8 | 24.2 |
| Other | 102 | 24.2 | 100.0 |
| *Driving Experience* | | | |
| 0–5 years | 78 | 18.5 | 18.5 |
| 6–15 years | 128 | 30.4 | 48.9 |
| 16+ years | 215 | 51.1 | 100.0 |
| *Car Type* | | | |
| Basic Personal Car | 382 | 90.7 | 90.7 |
| Other | 39 | 9.3 | 100.0 |
| *Age of Car* | | | |
| 0–3 year | 170 | 40.4 | 52.0 |
| 3–10 years | 212 | 50.4 | 90.7 |
| 10+ years | 39 | 9.3 | 100.0 |
| *Car Usage Period* | | | |
| Every hour of the day | 137 | 32.5 | 32.5 |
| Weekdays home-work | 201 | 47.7 | 80.3 |
| Weekends only | 83 | 19.7 | 100.0 |
| **Total** | **421** | **100.0** | |

### 2.2. Survey Questions and Materials

The survey questionnaire consisted of three subsections. The first set of questions was about gender, age, education level, profession, city living, driving experience (in terms of years of driving), car type, car age, and car usage period within a week. The second set of questions was about the driving expressions the participants had experienced (faced) in traffic; the driving expressions that the participants had themselves reflected (used) or witnessed in traffic; and whether or not they knew their legal rights and responsibilities in traffic. The third set of questions was asked to measure driving anger based on a 14-item short form of the Driving Anger Scale (DAS), which was developed by Deffenbacher et al. [11].

**a.　Short Form Scale of the Driving Anger Scale (DAS)**

The original Driving Anger Scale was developed by Deffenbacher et al. from the data of 1500 college students in the USA [11]. A cluster analysis of responses from potentially anger-inducing driving-related situations yielded a 33-item driving anger scale (DAS) ($\alpha = 0.90$) which possessed six reliable subscales called "Hostile Gestures", "Illegal Driving", "Police

Presence", "Slow Driving", "Discourtesy", and "Traffic Obstructions". The Cronbach's alpha coefficients of the subscales ranged from 0.78 to 0.87. A 14-item short scale ($\alpha$ = 0.80) was developed from scores that were highly correlated (r = 0.95) with the scores on the 33-item form. This utilized a five-point Likert scale with a score for each item ranging from 1 to 5 (i.e., 1 = not at all; 5 = very much).

In this study, one of the original short scale items, namely "Someone runs a red light or stop sign", was replaced with "Someone flashes the bright about your driving". This was done as it is thought that the latter type of event occurs more frequently in anger-eliciting situations in local traffic (i.e., Turkey) than the former.

**b.    Driving Anger Expression Related Survey Questions**

The Driving Anger Expression Inventory (DAX) scale was developed by Deffenbacher et al. [18]. The 49-item DAX scale assesses how people express their anger while driving. In this study, the four subscales of expressing anger while driving were factored from the DAX. These subscales are: (1) a 12-Item "Verbal Aggressive Expression" ($\alpha$ = 0.88) which assesses the verbally aggressive expressions of anger; (2) an 11-item "Personal Physical Aggressive Expression" ($\alpha$ = 0.81) assessing the ways in which the person uses themself to express their anger; (3) an 11-item "Use of the Vehicle to Express Anger" ($\alpha$ = 0.86), i.e., assessing the ways in which the person uses their vehicle to express their anger; and (4) a 15-item "Adaptive/Constructive Expression" ($\alpha$ = 0.90) assessing the ways in which the person copes positively with their anger.

In this study, to evaluate the participants' driving anger expressions, similar groupings were used as in the original DAX scale forms. In addition, "Honking for Aggressive Expression" was added as an additional item, as it is thought to be one of the most frequently used anger expressions in local traffic. In addition, to evaluate the driving anger expressions that were experienced (faced) by the participants personally in traffic, similar groupings were used as in the original DAX scale forms.

**3. Results**

*3.1. Driving Anger Expressions Reflected and Experienced as per Gender*

The highest difference in the type of anger reflection that is used more frequently by females than by males was found in "Grumble in a way that the other party will not hear" (66.4% vs. 53.7%). On the other hand, males reflected their anger more frequently than females via "Physical violence", "Turning on the high beams", and "Intimidating by tailgating" (4.5%, 19.4%, 12.9% vs. 2.3%, 10.9%, 8.6%, respectively). The other five types of anger expressions were selected by both genders within the 20% range difference.

The frequency of the selections with respect to the multiple choice question of "What are the most frequent driving anger expressions reflected (used) by you in traffic?", as categorized by gender in terms of % of cases, is shown in Figure 2a.

Based on this, the two most frequently preferred anger expressions were: "Honking" (70.5%) and "Grumble in a way that the other party will not hear" (60.3%). Females (53.2% and 57.5%) had higher frequencies than males (46.8% and 42.5%) in both of these cases. The two least frequently preferred anger expression types were: "Physical violence" (3.3%) and "Blocking the car" (5.9%). Males (64.3% and 52.0%) had higher frequencies than females (35.7% and 48.0%) in both of these cases.

The largest difference in anger type that was experienced by females more frequently than males was found in: "Drive up close to bumper" (71.8% vs. 55.7%). On the other hand, the anger expressions more frequently experienced by males than females were: "Physical violence", "Turning on the high beams", and "Cursing" (4.5%, 46.8%, 32.3% vs. 0.9%, 35.0%, 24.5%, respectively). The other five types of anger expression experienced were selected by both genders within the 5% range difference.

The frequency of the selections with respect to the multiple-choice question of "What are the most frequent driving anger expressions experienced (faced) by you in traffic?", as categorized by gender in terms of % of cases, is shown in Figure 2b.

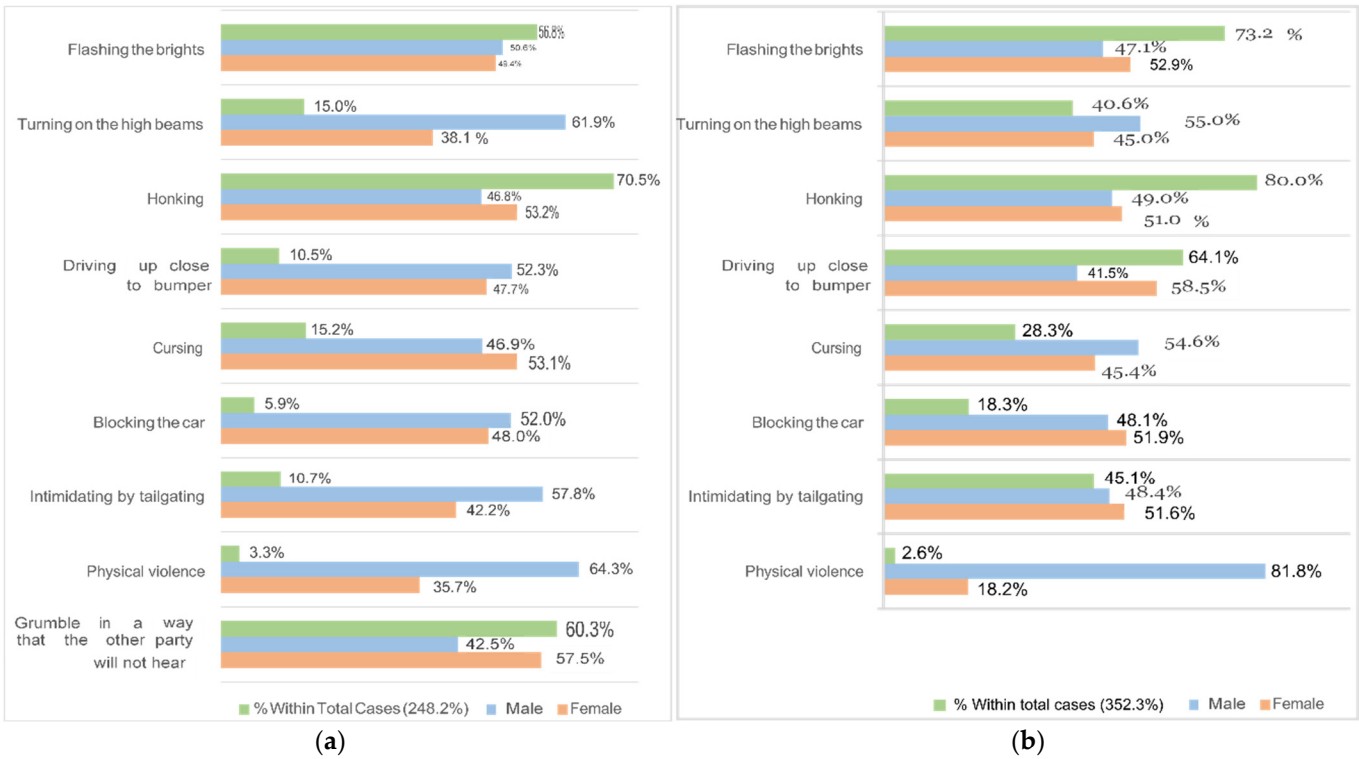

**Figure 2.** (**a**) Frequency of the driving anger expressions reflected (used) by the driver by gender in terms of % of cases; (**b**) Frequency of the driving anger expressions experienced (faced) by the driver by gender in terms of % of cases.

Based on this, the three most frequently experienced anger expressions were "Honking" (80.0%), "Flashing the bright" (73.2%), and "Driving up close to bumper" (64.1%). Females (51.0%, 52.9%, and 58.5%, respectively) had higher frequencies than males (49.0%, 47.1%, and 41.5% respectively) in all three of these cases. The two least frequently experienced anger expression types were: "Physical violence" (2.6%) and "Blocking the car" (18.3%). Males (81.8%) had a considerably higher frequency than females (11.2%) in the case of "Physical violence", while females (51.9%) had a slightly higher frequency than males (48.1%) in the "Blocking the car" case.

### 3.2. Comparisons among Different Country DAS Means

Table 2 shows the mean scores of the DAS from the studies of various countries, both for the 33-item long as well as for the various number of short item forms, where applicable. In this study, the three highest subscale means for the Turkish sample were found in: "Hostile Gestures (M = 3.0, SD = 0.99)", "Illegal Driving (M = 3.0, SD = 1.20)", and "Discourtesy (M = 2.9, SD = 0.86)"; and the lowest one was "Police Presence (M = 1.9, SD = 0.94)". These findings were in agreement with the findings of Yasak et al. [19] for Turkish drivers, Lajunen et al. [20] for UK drivers, and Sullman et al. [17] for NZ drivers. In general, in the USA, Deffenbacher et al. [11]—and in Malaysia, Kamarudin et al. [16]— reported the highest DAS scores. In this study, the mean DAS score for the total short form was M = 2.65, SD = 0.73. This finding was comparable to the UK and NZ findings. There was a difference in the mean scores for "Police Presence" between Turkish and American drivers (M = 1.9; M = 3.0), as also reported previously by Yasak et al. [20]. There was also a difference in the mean scores for "Discourtesy" between Turkish and American drivers (M = 2.9; M = 3.9).

**Table 2.** DAS means for USA [11], UK [21], NZ [18], Malaysia [17], Turkey [20], and Turkey (for this study).

| Subscales of DAS | USA 1994 | | UK 1998 | | NZ 2006–2013 | | Malaysia 2017 | | Turkey 2009 | | Turkey This Study | |
|---|---|---|---|---|---|---|---|---|---|---|---|---|
| | No. of Items | M | No. of Items | M | No. of Items | M | No. of Items | M | No. of Items | M | No. of Items | M (SD) |
| Discourtesy | 9 | 3.9 | 9 | 2.7 | 9 | 3.5 | 4 | 3.8 | 9 | 3.6 | 3 | 2.9 (0.86) |
| Traffic Obstruction | 7 | 3.3 | 7 | 2.0 | 7 | 2.7 | 4 | 3.2 | 7 | 3.1 | 3 | 2.5 (0.96) |
| Hostile Gesture | 3 | 3.2 | 3 | 2.3 | 3 | 2.7 | 4 | 3.5 | 3 | 3.4 | 3 | 3.0 (0.99) |
| Slow Driving | 6 | 3.2 | 6 | 2.0 | 6 | 2.8 | 4 | 3.3 | 6 | 2.9 | 2 | 2.5 (0.90) |
| Police Presence | 4 | 3.0 | 4 | 1.4 | 4 | 1.9 | 4 | 2.1 | 4 | 2.2 | 2 | 1.9 (0.94) |
| Illegal Driving | 4 | 2.7 | 4 | 2.3 | 4 | 3.3 | 4 | 3.3 | 4 | 3.5 | 1 | 3.0 (1.20) |
| DAS Total (33 item) | 33 | 3.2 | 33 | 2.1 | 33 | 2.8 | - | - | 33 | 3.1 | | |
| DAS Total | 14 | 3.4 | 21 | 2.4 | 14 | 2.7 | 24 | 3.2 | - | - | 14 | 2.65 (0.73) |

### 3.3. Factor Analysis

The short form DAS with 14 items was applied, with the exception of replacing one item in the original scale, to participants from the Turkish population. In this study, one of the original short form items, namely "Someone runs a red light, or stop sign (Illegal Driving subscale)" was replaced with "Someone flashes the bright about your driving (considered in Hostile Gesture subscale)"; as the latter is thought to be more frequently used in anger-eliciting situations locally. To find out the reliability and validity of the scale and to reduce the data to subscales, the data obtained were subjected to a principal axis factor analysis with oblique rotation, since there were relatively large factor intercorrelations (0.306, 0.399, 0.546, 0.378, and 0.437). The comparisons to the original DAS scale were developed by Deffenbacher et al. [11] and are detailed in the conclusions section. The correlation matrix showed that many of the item coefficients were above 0.3, with a determinant value of 0.003. The Kaiser–Meyer–Olkin value was 0.89, thus exceeding the recommended value of 0.6 [22,23]; and Bartlett's test of sphericity [24] was established with statistical significance ($p < 0.001$), thereby supporting the factorability of the correlation matrix. Table 3 shows the mean scores, the SDs of the total and of each DAS item, as well as the reliability of the subscales (Cronbach's alpha).

The factors with eigenvalues that were greater than 0.7 were retained based on Jolliffe's [25,26]. The factor analysis results showed that a total of five factors in combination explained 71.43% of the variance ($\alpha = 0.88$). Factor 1, called "Impatience", consisted of three items; one "Slow Driving", one "Traffic Obstruction", and one "Discourtesy" item; accounted for the variance of 40.53%, and had an eigenvalue of 5.67 ($\alpha = 0.74$). Factor 2, called "Discourtesy", consisted of four items; one "Illegal Driving", one "Slow Driving", and two "Discourtesy" items, accounted for 11.89% of the variance, and possessed an eigenvalue of 1.67 ($\alpha = 0.73$). Although item 6 (Someone speeds up when you try to pass him/her) was loaded on factor 4, based on its cultural perception and the reliable $\alpha$ values of the factor analyses, it was included in "Discourtesy" instead. Factor 3, called "Hostile Gesture", consisted of three items; two "Hostile Gesture" items, and one new item called "Someone flashes the bright about your driving", accounted for 7.17% of the variance, and possessed an eigenvalue of 1.00 ($\alpha = 0.78$). Factor 4, called "Police Presence" consisted of two "Police Presence" items, accounted for 6.15% of the variance, and possessed an eigenvalue of 0.86 ($\alpha = 0.75$). Factor 5, called "Traffic Obstruction", consisted of two "Traffic Obstruction" items, accounted for 5.69% of the variance, and possessed an eigenvalue of 0.80 ($\alpha = 0.70$).

**Table 3.** Mean scores, the SDs of the total and of each DAS item, as well as the reliability of the subscales (Cronbach's α).

| Item No. | Item Causing Driving Anger | Mean | SD |
|---|---|---|---|
| | *Impatience (α = 0.74)* | *2.30* | *0.91* |
| 7 | Someone is slow in parking and is holding up traffic | 2.14 | 1.10 |
| 8 | You are stuck in a traffic jam | 2.46 | 1.16 |
| 11 | A cyclist is riding in the middle of the lane and is slowing traffic | 2.30 | 1.11 |
| | *Discourtesy (α = 0.73)* | *3.07* | *0.85* |
| 1 | Someone is weaving in and out of traffic | 3.00 | 1.20 |
| 2 | A slow vehicle on a mountain road will not pull over and let people by | 2.88 | 1.10 |
| 3 | Someone backs right out in front of you without looking | 3.52 | 1.09 |
| 6 | Someone speeds up when your try to pass him/her | 2.86 | 1.19 |
| | *Hostile Gesture (α = 0.78)* | *3.02* | *0.99* |
| 4 | Someone flashes the brights about your driving | 2.75 | 1.13 |
| 9 | Someone makes an obscene gesture toward you about your driving | 3.45 | 1.28 |
| 10 | Someone honks at you about your driving | 2.85 | 1.15 |
| | *Police Presence (α = 0.75)* | *1.88* | *0.94* |
| 5 | You pass a radar speed trap | 2.14 | 1.13 |
| 12 | A police officer pulls you over | 1.62 | 0.96 |
| | *Traffic Obstruction (α = 0.70)* | *2.58* | *1.07* |
| 13 | A truck kicks up sand or gravel on the car you are driving | 2.88 | 1.26 |
| 14 | You are driving behind a large truck and you cannot see around it | 2.29 | 1.17 |
| | *DAS Total (α = 0.88)* | *2.65* | *0.73* |

### 3.4. The Correlations between DAS and Its Subscales

Table 4 shows the Pearson correlation coefficients of the DAS with its subscales. Results showed that there were significant and positive correlations (ranging from r = 0.31 ($p < 0.001$) to r = 0.61 ($p < 0.001$)) between the subscales of the DAS. There were also significant and positive correlations (ranging from r = 0.65 ($p < 0.001$) to r = 0.821 ($p < 0.001$)) between the subscales and DAS total score.

**Table 4.** Pearson correlations of the subscales of DAS.

| Subscales of DAS | 1 | 2 | 3 | 4 | 5 | 6 |
|---|---|---|---|---|---|---|
| 1. Impatience | - | 0.502 ** | 0.528 ** | 0.547 ** | 0.569 ** | 0.811 ** |
| 2. Discourtesy | | - | 0.606 ** | 0.366 ** | 0.488 ** | 0.817 ** |
| 3. Hostile Gesture | | | - | 0.311 ** | 0.454 ** | 0.790 ** |
| 4. Police Presence | | | | - | 0.504 ** | 0.651 ** |
| 5. Traffic Obstruction | | | | | - | 0.752 ** |
| 6. DAS Total | | | | | | - |

** $p < 0.001$.

A multivariate analysis of variance (MANOVA) revealed that driving experience had a significant main multivariate effect on DAS subscale scores combined with the dependent variables (F (10, 830) = 4.41, $p < 0.001$; Pillai's Trace = 0.10; partial eta squared = 0.05). Given the significance of the overall test, the main univariate effects were examined using Bonferroni's adjusted alpha level of 0.01.

### 3.5. Effects of Demographics and Various Independent Variables on DAS and Its Subscales

The effects of gender, age, education level, profession, city living, driving experience, car type, car age, car usage period, legal rights, and awareness of responsibilities on the total DAS and its subscales were investigated.

MANOVA revealed that age had a significant main multivariate effect on the DAS subscale scores combined with the dependent variables (F (20, 1660) = 2.99, *p* < 0.001; Pillai's Trace = 0.14; partial eta squared = 0.035). Given the significance of the overall test, the main univariate effects were examined using Bonferroni's adjusted alpha level of 0.01. Significant main univariate effects for age were obtained for "Discourtesy" (F (4, 416) = 3.66, *p* < 0.01, partial eta square = 0.034); "Hostile Gesture" (F (4, 416) = 5.58, *p* < 0.001, partial eta square = 0.051); and for "Traffic Obstruction" (F (4, 416) = 6.48, *p* < 0.001, partial eta square = 0.059). Significant age pairwise differences were investigated using Scheffe's post hoc tests. The participants who were aged between 18–25 years (M = 14.58, SD = 3.12) had significantly higher mean scores of "Discourtesy" than those of 61+ years (M = 10.68, SD = 4.05, *p* < 0.05). The participants aged between 18–25 years (M = 10.75, SD = 3.28) had significantly higher mean scores of "Hostile Gesture" than those of 61+ years (M = 7.50, SD = 2.61, *p* < 0.05). Lastly, those who were aged between 26–40 years (M = 9.59, SD = 2.81) had significantly higher mean scores than those of 61+ years (M = 7.50, SD = 2.61, *p* < 0.05). The participants aged between 18–25 years (M = 7.75, SD = 2.05) had the significantly higher mean scores of "Traffic Obstruction" than those of all other age groups, which were between 26–40 years (M = 5.16, SD = 2.12, *p* < 0.01), 41–50 years (M = 5.17, SD = 2.09, *p* < 0.01), 51–60 years (M = 5.10, SD = 2.06, *p* < 0.01), and 61+ years (M = 3.95, SD = 1.86, *p* < 0.001). No other significant pairwise mean differences were investigated for the age groups on the DAS subscale scores.

A one-way between-groups analysis of variance (ANOVA) revealed that age had a significant univariate effect on the DAS total scores. Further, there was a statistically significant difference at *p* < 0.001 on the DAS total scores for the five age groups (F (4, 416) = 4.71, *p* < 0.001, $\omega2$ = 0.034). Significant age pairwise differences were investigated using Scheffe's post hoc tests. Based on this, participants aged between 18–25 years (M = 46.25, SD = 8.07) had significantly higher DAS total scores than those aged between 41–50 years (M = 36.89, SD = 10.75, *p* < 0.05); as well as those who were between the ages of 51–60 years (M = 35.20, SD = 9.84, *p* < 0.05); and those who were of 61+ years (M = 32.23, SD = 10.62, *p* < 0.01).

Table 5 shows the summary of the factor analysis results for the short DAS questionnaire.

Significant main univariate effects for driving experience were obtained for "Discourtesy" (F (2, 418) = 9.05, *p* < 0.001, partial eta square = 0.042); "Hostile Gesture" (F (2, 418) = 8.56, *p* < 0.001, partial eta square = 0.039); and for "Traffic Obstruction" (F (2, 418) = 7.96, *p* < 0.001, partial eta square = 0.037). Significant driving experience pairwise differences were obtained using Scheffe's post hoc tests. Based on this, participants with driving experience of 0–5 years (M = 13.05, SD = 3.06) had significantly higher scores of "Discourtesy" than those with driving experience of 16+ years (M = 11.58, SD = 3.52, *p* < 0.01). In addition, those who had 6–15 years driving experience (M = 12.91, SD = 3.21) had significantly higher scores of "Discourtesy" than those with driving experience of 16+ years (M = 11.58, SD = 3.52, *p* < 0.01). The participants with driving experience of 0–5 years (M = 9.77, SD = 2.84) had significantly higher scores of "Hostile Gesture" than those with driving experience of 16+ years (M = 8.48, SD = 3.10, *p* < 0.01). Furthermore, those with 6–15 years of driving experience (M = 9.58, SD = 2.64) had significantly higher scores of "Hostile Gesture" than those with driving experience of 16+ years (M = 8.48, SD = 3.10, *p* < 0.01). The participants with driving experience of 0–5 years (M = 5.81, SD = 2.29) had significantly higher scores of "Traffic Obstruction" than those with driving experience of 16+ years (M = 4.79, SD = 2.00, *p* < 0.01). Moreover, those with 6–15 years driving experience (M = 5.41, SD = 2.15, *p* < 0.01) had significantly higher scores of "Traffic Obstruction" than those with driving experience of 16+ years (4.79, SD = 2.00, *p* < 0.05). No other significant pairwise mean differences were investigated for the different driving experience groups on the DAS subscale scores.

ANOVA revealed that driving experience had a significant univariate effect on the DAS total scores. There was a statistically significant difference at *p* < 0.001 on the DAS total scores for the three driving experience groups (F (2, 418) = 7.35, *p* < 0.001, $\omega2$ = 0.029). Significant driving experience pairwise differences were investigated using

Scheffe's post hoc tests. Based on this, participants with driving experience of 0–5 years (M = 39.04, SD = 9.67) had significantly higher scores of the DAS total than those of 16+ years (M = 35.30, SD = 10.70, $p < 0.01$). In addition, those with driving experience of 6–15 years (M = 39.05, SD = 9.00) had significantly higher scores of the DAS total than those of 16+ years (M = 35.30, SD = 10.70, $p < 0.01$).

**Table 5.** Summary of factor analysis results for the short DAS questionnaire factor.

| Item No. | Item Description | Factor | | | | | |
|---|---|---|---|---|---|---|---|
| | | (1) | (2) | (3) | (4) | (5) | |
| | | Impatience | Discourtesy | Hostile Gesture | Police Presence | Traffic Obstruction | Communalities |
| 7 | Someone is slow in parking and is holding up traffic | **0.801** | −0.032 | −0.049 | 0.030 | −0.006 | 0.681 |
| 11 | A bicyclist is riding in the middle of the lane and is slowing traffic | **0.372** | 0.024 | −0.150 | 0.134 | 0.175 | 0.427 |
| 8 | You are stuck in a traffic jam | **0.370** | 0.102 | −0.149 | 0.180 | 0.097 | 0.458 |
| 3 | Someone backs right out in front of you without looking | −0.017 | **0.793** | −0.033 | 0.087 | 0.001 | 0.699 |
| 2 | A slow vehicle on a mountain road will not pull over and let people by | 0.204 | **0.669** | 0.154 | −0.057 | 0.212 | 0.577 |
| 1 | Someone is weaving in and out of traffic | −0.110 | **0.608** | −0.136 | −0.024 | −0.053 | 0.413 |
| 10 | Someone honks at you about your driving | 0.108 | −0.044 | **−0.820** | 0.014 | 0.033 | 0.741 |
| 9 | Someone makes an obscene gesture toward you about your driving | 0.015 | 0.084 | **−0.746** | −0.096 | 0.097 | 0.635 |
| 4 | Someone flashes the brights about your driving | 0.109 | 0.318 | **−0.340** | 0.170 | −0.150 | 0.453 |
| 5 | You pass a radar speed trap | 0.037 | 0.032 | 0.061 | **0.815** | −0.047 | 0.648 |
| 12 | A police officer pulls you over | 0.034 | −0.075 | 0.024 | **0.707** | 0.173 | 0.619 |
| 6 | Someone speeds up when you try to pass him/her | 0.265 | 0.167 | −0.163 | **0.313** | −0.031 | 0.452 |
| 14 | You are driving behind a large truck and you cannot see around it | 0.200 | 0.064 | −0.067 | 0.047 | **0.665** | 0.718 |
| 13 | A truck kicks up sand or gravel on the car you are driving | −0.097 | 0.089 | −0.194 | 0.251 | **0.460** | 0.473 |
| | Eigenvalues | 5.67 | 1.67 | 1.00 | 0.86 | 0.80 | |
| | % of variance | 40.53 | 11.89 | 7.17 | 6.15 | 5.69 | |
| | Cronbach's α | 0.74 | 0.73 | 0.78 | 0.75 | 0.70 | |

Note: Factor loadings over 0.30 shown in bold.

MANOVA revealed that the car usage period had a significant main multivariate effect on the DAS subscale scores combined with the dependent variables (F (10, 830) = 3.84, $p < 0.001$; Pillai's Trace = 0.088; partial eta squared = 0.044). Given the significance of the overall test, the main univariate effects were examined using Bonferroni's adjusted alpha level of 0.01. Significant main univariate effects for car usage periods were obtained for "Traffic Obstruction" (F (2, 418) = 9.23, $p < 0.001$, partial eta square = 0.042). The significant car usage period pairwise differences were obtained using Scheffe's post hoc tests. Based on this, the participants with a car usage period of weekends only (M = 5.94, SD = 2.11) had significantly higher scores of "Traffic Obstruction" than those with a car usage period of every hour of the day (M = 4.69, SD = 2.21, $p < 0.001$), and those with a car

usage period on the weekdays between home and work (M = 5.17, SD = 2.01, $p < 0.05$). The significant main univariate effects for car usage period were obtained for "Impatience" with a homogeneity of variances not being assumed (Welch's F (2, 194.51) = 4.86, $p < 0.017$ and Brown–Forsythe F (2, 275.51) = 4.92, $p < 0.017$, partial eta square =0.025). The significant car usage period pairwise differences were obtained using Games–Howell post hoc tests. Based on this, the participants with a car usage period of weekends only (M = 7.75, SD = 3.00) had significantly higher scores of "Impatience" than those with a car usage period of weekdays between home and work (M = 6.60, SD = 2.35, $p < 0.01$). No other significant pairwise mean differences were investigated among car usage period groups on the DAS subscale scores.

ANOVA revealed that car usage period had a significant univariate effect on the DAS total scores. There was a statistically significant difference at $p < 0.05$ on the DAS total scores for the three car usage period groups. Since the homogeneity of variances were not assumed, Welch's F was (2, 199.73) = 4.27, and $p < 0.05$, $\omega 2 = 0.015$. The significant car usage period pairwise differences were investigated using Games–Howell post hoc tests. Based on this, the respondents with a car usage period of weekends only (M = 39.94, SD = 10.55) had significantly higher scores of the DAS total than those whose car usage period was every hour of the day (M = 35.56, SD = 11.22, $p < 0.05$).

The multivariate and univariate analyses of the variance for gender, education level, profession, city living, car type, car age, legal rights, and awareness of responsibilities on the total DAS and on its subscales did not reveal significant effects.

### 3.6. The Relationship between DAS and Gender

The reported DAS mean scores for the five subscales and DAS total, by gender, were 220 female (52.3%) and 201 male (47.7%) participants. The five anger subscales did not show significant differences in the means between females and males. However, female participants reported more anger provoked on all subscales except "Police Presence" than in the males. Females also reported a higher overall mean level of DAS score than males.

### 3.7. The Relationship between Driver Anger Expressions and Gender

In this study, "Personal Physical Aggressive Expression" was removed from the analysis since it had only one case. Our sample of 219 female (52.1%) and 201 male (47.9%) drivers in a chi-square test showed that there was a significant association between driver anger expressions and gender ($\chi 2$ (3, N = 420) = 16.50, $p < 0.001$, Cramer's V = 0.20). A post hoc test was performed and, in order to reduce an inflated Type I error, Bonferroni's correction with $p < 0.0063$ was used. The results showed that the significance was driven by the "Using the Vehicle" and "Adaptive/constructive Anger Expressions" subscales of the drivers. Females were observed (20.5%) significantly less than expected, whilst males were observed (35.3%) significantly more than expected, with regard to expressing their anger by using the vehicle. Based on the odds ratio, males were nearly 2 times more likely to use a vehicle for aggressive anger expression than females (OR = 1/0.47 = 2.11). Conversely, females were observed (39.7%) significantly more than expected, whilst males were observed (26.9%) significantly less than expected with regard to expressing their anger with adaptive/constructive expressions. Based on the odds ratio, females were nearly 2 times more likely to use adaptive/constructive expressions than males (OR = 1.79). Verbal aggressive expression was reported as the least preferred anger expression by both genders and was thus deemed non-significant. In addition, honking for the purposes of aggressive expression was not observed as significantly different than was expected.

### 3.8. The Relationship between Being a Lawyer and Knowing Legal Responsibilities and Rights in Traffic Violence Situations

With our sample of 31 lawyers (7.4%) and 390 other profession (92.6%) drivers, a chi-square test showed that there was a significant association (relation) between being a lawyer and knowing ones' legal responsibilities and rights in traffic violence situations ($\chi 2$(1, N = 421) = 55.47, $p < 0.001$, Cramer's V = 0.36). Lawyers were observed (80.6%) signif-

icantly higher than expected, whilst other professions were observed (20.5%) significantly less than expected, with respect to knowing one's legal rights and responsibilities in traffic violence situations. Based on the odds ratio, lawyers were nearly 16 times more likely to know their legal rights and responsibilities in traffic violence situations than those in other professions (OR = 16.15).

*3.9. The Impact of Various Categorical Predictors on Reporting Low Driving Anger Scores*

Direct logistic regression was performed to assess the impact of several factors and the likelihood of whether or not participants would report that they had a low driving anger score. The low driving anger score cut-off level was calculated as the 1st quartile of the survey data on the DAS total scores. Based on this, the scores below 30 were accepted as a low driving anger level. There was only one survey answer with "Personal Physical Aggressive Expression", hence it was excluded from the analysis. In the analysis, 102 (24.3%) low scores and 318 (75.7%) non-low scores were used. In addition, age as a categorical variable was not included in this analysis, since all the 18–25 years old participants had answered within the low anger score level range (N = 12, 2.9%). Therefore, age was analyzed separately. The model was composed of one dependent variable (LDAS; not low = 0, low = 1), and nine independent categorical variables in the low driving anger scale, namely: gender (male = 0, female = 1); education level (<university = 0, university = 1); traffic density based on driving in Istanbul with heavy traffic (light = 0, heavy = 1); driving experience (0–5 years = 0, 6–15 years = 1, 16+ years = 2); car type (basic personal = 0, other = 1); car age (0–3 years = 0, 3–10 years = 1, 10+ years = 2); driver anger expression (using the vehicle = 0, honking = 1, verbal aggression = 2, adaptive/constructive = 3); car usage period (every hour of the day = 0, weekdays between home and work = 1, weekends only = 2); and the awareness of legal rights and responsibilities (no = 0, yes = 1). The reference categories were coded with 0.

As summarized in Table 6, the full model containing all predictors was statistically significant ($\chi2$ (14, N = 420) = 46.38, $p < 0.001$), hence the model was able to distinguish between participants who reported and who did not report themselves to have low driving anger. The model correctly classified 75.7% of cases. Only three of the independent variables made a statistically unique and significant contribution to the model (education level, driving experience, and awareness of legal rights and responsibilities). The strongest categorical predictor variable that predicted a low anger scale was education level, with an odds ratio of 3.15, $p < 0.05$. This indicated that participants who were not university graduates were nearly 3.2 (=1/0.32) times more likely to report a low driving anger level than those who were university graduates, even when controlling for all other factors in the model. The driving experience categorical predictor variable that predicted a low anger scale was with an odds ratio of 3.11, $p < 0.01$. This indicated that participants who had been driving for over 16 years were nearly 3 times more likely to report a low driving anger level than those who were driving less than 5 years, even when controlling for all other factors in the model. The legal rights and awareness of responsibilities categorical predictor variable that predicted a low anger scale was with an odds ratio of 2.48, $p < 0.001$. This indicated that participants who were aware of their legal rights and responsibilities were nearly 2.5 times more likely to report low driving anger than those who were not, even when controlling for all other factors in the model.

**Table 6.** Various independent variables logistic regression predicting likelihood of low driving anger.

| | B | S.E. | Wald | df | *p* | Odds Ratio | 95% C.I. Lower | 95% C.I. Upper |
|---|---|---|---|---|---|---|---|---|
| Gender (Female) | 0.33 | 0.27 | 1.55 | 1 | 0.213 | 1.40 | 0.83 | 2.37 |
| Education Level (University) | −1.15 | 0.48 | 5.78 | 1 | 0.016 | 0.32 | 0.12 | 0.81 |
| Traffic Density (Heavy) | −0.40 | 0.29 | 1.98 | 1 | 0.160 | 0.67 | 0.38 | 1.17 |

**Table 6.** *Cont.*

| | B | S.E. | Wald | df | *p* | Odds Ratio | 95% C.I. Lower | 95% C.I. Upper |
|---|---|---|---|---|---|---|---|---|
| Driving Experience | | | 10.51 | 2 | 0.005 | | | |
| 6–15 years | 0.46 | 0.43 | 1.13 | 1 | 0.288 | 1.58 | 0.68 | 3.65 |
| 16+ years | 1.13 | 0.40 | 8.10 | 1 | 0.004 | 3.11 | 1.42 | 6.80 |
| Car Type (Non-basic personal) | −0.83 | 0.46 | 3.28 | 1 | 0.070 | 0.43 | 0.18 | 1.07 |
| Car Age | | | 1.84 | 2 | 0.399 | | | |
| 3–10 years | −0.32 | 0.26 | 1.47 | 1 | 0.225 | 0.73 | 0.44 | 1.22 |
| 10+ years | −0.44 | 0.47 | 0.90 | 1 | 0.343 | 0.64 | 0.26 | 1.60 |
| What is the most frequent driving anger expression reflected by you while driving? | | | 3.43 | 2 | 0.180 | | | |
| Honking for Aggressive Expression | −0.35 | 0.27 | 1.62 | 1 | 0.203 | 0.71 | 0.41 | 1.21 |
| Verbal Aggressive Expression | −0.66 | 0.38 | 3.02 | 1 | 0.082 | 0.51 | 0.24 | 1.09 |
| Adaptive/Constructive Expression | | | 3.04 | 3 | 0.385 | | | |
| Car usage period | 0.04 | 0.31 | 0.02 | 1 | 0.884 | 1.05 | 0.57 | 1.91 |
| Weekdays between home and work | −0.14 | 0.57 | 0.06 | 1 | 0.805 | 0.87 | 0.29 | 2.64 |
| Weekends only | −0.45 | 0.32 | 1.94 | 1 | 0.163 | 0.64 | 0.34 | 1.20 |
| Under traffic violence situation, do you know your legal responsibilities and rights? (Yes) | 0.91 | 0.27 | 11.57 | 1 | 0.001 | 2.48 | 1.47 | 4.19 |
| Constant | −0.31 | 0.70 | 0.19 | 1 | 0.659 | 0.74 | | |

Note: 0.10 (Hosmer and Lemeshow), 0.10 (Cox and Snell), and 0.16 (Nagelkerke). Model $\chi2$ (14, N = 420) = 46.38, $p < 0.001$.

### 3.10. The Impact of an Age Categorical Predictor on Reporting Low Driving Anger

Direct logistic regression was performed to assess the impact of age and the likelihood that participants would report that they had a low driving anger score or not. The model was composed of one independent categorical variable, namely age (26–40 years = 0, 41–50 years = 1, 51–60 years = 2, 61+ years = 3). The reference category was coded with 0. Regarding the 18–25 years old age category, it was not entered into the analysis since they all reported themselves with a low driving anger score. In the analysis, 103 (25.2%) low scores and 306 (74.8%) non- low scores were used. As summarized in Table 7, the full model containing all predictors was statistically significant ($\chi2$ (3, N = 409) = 8.52, $p < 0.05$), hence the model was able to distinguish between respondents who reported and who did not report themselves to have a low driving anger score. The model correctly classified 74.8% of cases. Only one of the coded independent variables made a unique statistically significant contribution to the model (61+ years), which was with an odds ratio of 3.38, $p < 0.01$. This indicated that participants who were above 61 years were nearly 3.5 times more likely to report low driving anger than those who were 26–40 years old, even when controlling for all other factors in the model.

**Table 7.** Age as an independent variable logistic regression predicting likelihood of low driving anger.

| | B | S.E. | Wald | df | Sig. | Odds Ratio | 95% C.I. Lower | 95% C.I. Upper |
|---|---|---|---|---|---|---|---|---|
| Age | | | 8.61 | 3 | 0.035 | | | |
| 41–50 years | 0.39 | 0.27 | 2.11 | 1 | 0.146 | 1.47 | 0.87 | 2.47 |
| 51–60 years | 0.61 | 0.33 | 3.34 | 1 | 0.068 | 1.83 | 0.96 | 3.51 |
| 61+ years | 1.22 | 0.47 | 6.83 | 1 | 0.009 | 3.38 | 1.36 | 8.42 |
| Constant | −1.40 | 0.18 | 58.15 | 1 | 0.000 | 0.25 | | |

Note: 0.03 (Hosmer and Lemeshow), 0.02 (Cox and Snell), and 0.03 (Nagelkerke). Model $\chi2$ (3, N = 409) = 8.52, $p < 0.05$.

## 4. Discussion

This study indicated that the short form of the DAS, with its 14 items, is a highly reliable scale and may be implemented for the Turkish population as well. The factor analysis of the DAS, when using data from the Turkish population, clumped on a five-factor solution, which had similarities to the six-factor original 33-item form developed by Deffenbacher et al. [11] that was used for drivers in the USA. The main similarities were found on the "Hostile Gesture", "Police Presence", and "Traffic Obstruction" subscales. The "Discourtesy" scale in this study had 4 items, with 2 of them being from the same subscale items of the original form by Deffenbacher et al. [11]. A total of 3 out of 4 items on the "Discourtesy" scale were similar to those in a study by Sullman et al. [17], who conducted their study on the New Zealand population. The "Impatience" scale in this study had 3 items with similarities to studies that were conducted by Sullman et al. [17] (NZ population), Lajunen et al. [20,26] (UK population), and Parker et al. [27] (UK, Finland, and the Netherlands populations). The "Impatience" scale in this study had items from the "Slow Driving", "Traffic Obstruction", and "Discourtesy" scales of the original form by Deffenbacher et al. [11]. It was thought that this clumping of the factors arose from the fact that perceptions about traffic in the two countries differ.

In this study, there were no significant effects of gender on the DAS and its subscales that were similar to what was observed by Lajunen et al. [20] for the British; by Yasak et al. [19] for the Turkish; by Sullman et al. [28] for the New Zealanders; or by Kamarudin et al. [13] for Malaysian drivers. Deffenbacher et al. [11] found that there were gender mean differences on "Illegal Driving, Slow Driving, Police Presence and Traffic Obstructions", with small effect sizes. However, there was no significant difference found for these categories on the overall short DAS scale.

This study showed that females reflected their driving anger more frequently via "Adaptive/Constructive Expressions" than males, whereas males reflected more frequently via "Physical Expressions" and "Using the Vehicle" than females. In addition, the two most frequently preferred anger expressions were "Honking" and "Adaptive/Constructive Expressions". Females had higher frequencies than males in both dimensions. It has been thought that honking is a relatively safer way of expressing anger than other types, and is thus less likely to end in physical confrontation. Therefore, it is thought that females prefer this way of anger expression more than males. These results are similar to the study of Parker et al. [27]. The two least frequently preferred anger expression types were: "Physical Expressions" and "Using the Vehicle". Males had higher frequencies than females in both dimensions. These findings had similarities to studies in the USA [11,12,29,30] and in Turkey [14].

Based on the single choice question to check the relation between gender, as well as the reflection of one's driving anger expression, this study showed that there was a significant difference in terms of gender in the "Using the Vehicle for Aggressive Expression" and "Adaptive/constructive Expression" dimensions. Males were nearly two times more likely to use "Vehicle for Aggressive Anger Expression" than females, while females were nearly 2 times more likely to use "Adaptive/constructive Expressions". Similar differences for females and males, i.e., significant aggressive expression differences between the genders, were reported in the USA [12,18,29,30] and in Turkey [14]. Similarly, in this study, verbal aggressive expression did not show a significant difference in terms of gender by the researchers in the USA [18,30] and in Turkey [14]. Regarding "Honking for Aggressive Expression", although non-significant, females were more likely to honk for aggressive expression than males. In contrast to what was observed here, in Israel [15] it was reported that males were nearly twice as likely to honk than females, with a significant difference.

Previous research conducted in the UK [20,24], Israel [15], Britain–Finland–Netherlands [28], NZ [17,28], Turkey [19], and Malaysia [16] has shown that driving anger varies with age. The multivariate and univariate analyses of variances in this study revealed that on the total DAS, and on three of the DAS subscale scores, age had significant combined and univariate effects ("Discourtesy", "Hostile Gesture", and "Traffic Obstruction"); as a reaction

to "Traffic Obstruction", the 18–25 years age group had significantly higher mean DAS scores than all other age groups. It was noticeable that none in the 18–25 years age range had a score in the low DAS level (1st quartile of total DAS). Logistic regression analysis has also shown that those who are 61+ years old are nearly 3.5 times more likely to report low driving anger than those who were 26–40 years old, even when controlling for all other factors in the model. The general trend was that as the participants got older, their mean scores were lower on the DAS than in the studies that were conducted in the UK [20,24], Israel [15], NZ [17], Canada [31], and Malaysia [16].

On the other hand, a study conducted in Turkey [19] did not find any significant effects, by age, on driving anger. The multivariate and univariate analyses of variances in this study revealed that on the total DAS and on three of the DAS subscales scores, driving experience had significant combined and univariate effects ("Discourtesy", "Hostile Gesture", and "Traffic Obstruction"). Logistic regression analysis has also shown that respondents who were driving for over 16 years are nearly 3 times more likely to report low driving anger than those who were driving for less than 5 years, even when controlling for all other factors in the model. The general trend was that as the participants' driving experience increases, they score lower on the DAS. Research that was performed in order to study the effects of driving experience on the DAS in NZ [17] and in Malaysia [16] agreed with this study. However, the study conducted in NZ [17] noted that its findings may be affected due to the fact that those drivers who had been driving for a long time were also older ($r = 0.97$, $p < 0.001$). Drivers with greater experience might have higher self-confidence and thus be less anxious about the aggressiveness of other drivers. In the study conducted in Turkey [20], the opposite was reported for the Turkish drivers.

The multivariate and univariate analyses of variances in this study revealed that on the total DAS, as well as on the "Impatience" and "Traffic Obstruction" dimensions of the DAS subscales, car usage period had significant combined and univariate effects. Pairwise analysis showed that weekends only usage had significantly higher scores of traffic obstruction than those with a car usage period of every hour of the day, and those of weekdays between home and work. Regarding the impatience subscale, weekends only had significantly higher scores than the weekdays between the home and work group. For the overall DAS, the weekends only car usage period group had significantly higher mean scores than the every hour of the day car usage period group. This might be due to the drivers being annoyed by the traffic obstructions and thus finding themselves in more impatient situations during weekends because they want to reach their intended destination as soon as possible without losing their valuable leisure time on the roads. In Israel [15], it was observed that the relative risk of aggressive driving decreased from rush hour travel to weekends. They also noted that as traffic congestion increases, the likelihood of aggressive driving increases. These findings are in line with the drivers of the weekend only group and the light traffic expectations found in this study. Due to traffic obstructions faced over the weekend, they might have scored on the high end of the DAS.

Logistic regression analysis has shown that drivers without a university degree (i.e., elementary, secondary, and high school graduates) are nearly 3.2 times more likely to report low driving anger than those who were university graduates, even when controlling for all other factors in the model. The study conducted in Turkey [19] found, in relation to such an effect regarding education on the DAS for Turkish drivers, it is mentioned that drivers without a university degree had higher scores on the DAS subscale "Slow Driving" [32,33]. In line with these findings, as in this study, it might be that as education level increases, so do the reactions to driving anger-eliciting situations due to discourtesy and hostility increase; and thus, this results in them not reporting themselves as on a low DAS level.

When the drivers were asked "Under traffic violence situations, do you know your legal responsibilities and rights?", logistic regression analysis showed that drivers who responded with "yes" to this question were nearly 3 times more likely to report low driving anger than those who responded with "no", even when controlling for all other factors in the model. In addition to that, and unsurprisingly, a significant association was found

between being a lawyer and knowing legal rights and responsibilities under traffic violence situations. Lawyers were 16 times more likely to know their legal rights and responsibilities in a traffic violence situation than the other professions among the participants. Based on a review of the literature, no DAS-related studies have been found that have analyzed the effects of lawyer proficiency, legal rights, and responsibility awareness. One interesting result from this study was that, although lawyers had a very high likelihood (as was expected in comparison to the other professions) of knowing their legal rights and responsibilities in traffic violence situations, they did not significantly report themselves to have a low DAS level. However, drivers who declared themselves with awareness of legal rights and responsibilities, did significantly report themselves to have a low DAS level.

To summarize, factor analysis showed that the short form of the DAS, with 14 items in five dimensions, was correlated with each subscale and with the overall DAS. This revised five-factor solution ("Impatience", "Discourtesy", "Hostile Gesture", "Police Presence", and "Traffic Obstruction") of the original long form of DAS, as developed by Deffenbacher et al. [11] for the USA sample, can be used reliably for Turkish drivers as well as for further research purposes. The study revealed no significant effects of gender on the DAS, but significant effects of age, driving experience, and car usage period. The general trend was that as the participants became older, their driving experience increased, and thus they scored lower on the DAS. In other words, there were negative correlations between driver age, driving experience, and the DAS. The drivers who used their cars on weekends only had higher mean scores than the group of drivers who drove every hour and the group of drivers who drove on weekdays between home and work. This might be due to drivers being annoyed by traffic obstructions during weekends as they want to reach their intended destination as soon as possible without losing their valuable leisure time on the roads. Drivers without a university degree were found to be significantly more likely to report low driving anger than those who were university graduates. In addition, a unique research question that was investigated in this study showed that drivers who knew their legal responsibilities and rights were found to be significantly more likely to report low driving anger than those who did not. Further analysis, as expected, indicated that lawyers were significantly more likely to know their legal rights and responsibilities in traffic violence situations than drivers of other professions.

Additionally, males were significantly were more likely to use "Vehicle for Aggressive Anger Expression" than females, whilst females were significantly more likely to use "Adaptive/constructive Expressions". These findings indicate that there are gender differences in aggression; male drivers are more aggressive than female drivers. These findings are in line with previous studies. Moreover, the two most frequently preferred driving anger expressions reflected were: "Honking" and "Adaptive/Constructive Expressions". Additionally, the driving anger expression type that females reported to have experienced (faced) more frequently than males was "Using the Vehicle"; while males reported "Physical Expressions" and "Verbal Expressions" as their most frequent expressions. However, the two most frequently experienced driving anger expressions for both genders were "Honking" and "Using the Vehicle".

In summary, this study has revealed that people who are educated, who know their legal responsibilities in traffic violence situations, and who are old and experienced with respect to driving, have lower scores on the DAS and are better able to control their anger in traffic.

As is widely known, the limitations of this study might be associated with it being solely based on a self-report type of data, which was obtained via online survey. However, since all participants were informed of their anonymity beforehand, their responses were expected to reflect sincere experiences.

## 5. Conclusions

Road rage and sustainability are two distinct issues, but they can be related in certain ways. Road rage refers to aggressive and angry behavior exhibited by drivers, which often

results in dangerous or violent situations. Sustainability, on the other hand, is the practice of meeting the needs of the present without compromising the ability of future generations to meet their own needs. One way that road rage and sustainability are related is through the impact of vehicle emissions on the environment. Aggressive driving behaviors, such as speeding, rapid acceleration, and braking, can lead to increased fuel consumption and emissions which contribute to air pollution and climate change. Encouraging more sustainable driving practices, such as reducing speed and avoiding aggressive behaviors, can help to reduce emissions and thus improve air quality.

Road rage is a type of aggressive driving behavior that can negatively impact an individual's well-being. It can lead to increased stress, anxiety, and even physical harm. To promote well-being while driving, it is important to remain calm and avoid engaging in aggressive behavior.

Clause 3 of the Sustainable Development Goals of the United Nations is about well-being; moreover, sub-clause 3.6 regulates traffic.

There is a sound body of scientific evidence behind road safety interventions. Adopting and enforcing legislation relating to important risk factors–speed, drunk driving, motorcycle helmets, seatbelts, and child restraints–has been shown to lead to reductions in road traffic injuries [34].

Aggressive driving and road age also threaten human health and wellbeing. Aggressive and reckless driving has been identified as one of the leading causes of transport injuries. In addition, it is associated with significant psychiatric distress, which indicates the necessity of interventions, at least for target groups [35,36].

Road rage includes aggressive behaviors, such as excessive speeding, tailgating, horn honking, traffic weaving, profanity, obscene gestures, headlight flashing, etc. [37]. To understand this phenomenon, researchers have examined aggressive driving in terms of several levels of variables, such as the sociocultural, situational, and personal [38].

While perpetrators of road rage are known to have substance misuse disorders, depression and anxiety symptoms, as well as cluster B personality traits, victims of road rage can end up having cases of depression and post-traumatic stress disorder [39].

Driver anger is one of the most significant factors affecting the mental, as well as physical, health of individuals. Thus, it has importance according to the right to health. Anger can be both a cause and a consequence of poor mental health. As is already known, many psychological processes underlie transport behavior and mobility choices, from: perception to attention; and decision-making to affective processes. Thus, it is important that the research explores and deepens the psychological determinants of sustainable transport [40]. Moreover, it is aimed to focus on the psychological aspects that are connected to the road rage that is faced during sustainable transport, and its relationship to individual and public well-being.

Consequently, it is important to reduce traffic anger and road rage for the purposes of improving well-being. By doing this, it will also allow us to fulfil the sustainable development goals of the United Nations.

The research indicates that there is a negative correlation between drivers' age and drivers' anger. Therefore, it is recommended to organize training sessions specifically for young drivers, perhaps even starting from elementary school stages, on how to control driving anger and what the consequences might be otherwise.

In summary, road rage is a type of aggressive driving that can lead to dangerous and violent situations on the road. It can have a negative impact on a person's overall well-being, as well as the well-being of others on the road. To reduce the risk of road rage, it is important to stay calm and patient while driving, to avoid making rude or aggressive gestures, and to give other drivers space to maneuver.

It is suggested that to control, regulate, and lessen traffic violence, safe driving culture must be popularized substantially via educatory applications in digital or classroom environments. Additionally, effective audits and deterrent regulations are also suggested to decrease driving anger and violence. Practicing mindfulness and stress management

techniques can also help to improve overall well-being and to reduce the likelihood of road rage.

Since our study has revealed that the drivers who are old and experienced are less prone to road rage, it is suggested to start the training sessions with young and less experienced people. It is also recommended that the training sessions should contain details regarding legal responsibilities and penal legislation, since the drivers who are educated and know their legal responsibilities in relation to traffic violence situations are less prone to road rage.

Further, it is suggested that future studies expand to a larger population, particularly with respect to investigating driving anger levels and the expressions of commercial vehicle drivers.

**Author Contributions:** Conceptualization Z.R.; methodology Z.R. and O.P.; validation Z.R.; formal analysis Z.R. and O.P.; investigation Z.R. and O.P.; resources Z.R.; data curation Z.R.; writing—original draft preparation Z.R.; writing—review and editing Z.R. and O.P.; visualization Z.R.; and project administration Z.R. and O.P. All authors have read and agreed to the published version of the manuscript.

**Funding:** This research received no external funding.

**Institutional Review Board Statement:** The study was conducted according to the guidelines of the Declaration of Helsinki and followed the European regulations for personal data management, as well as the Turkish regulations for personal data management. Ethical review and approval were waived for this study because data collection did not imply any risk to participants and did not include biological measures.

**Informed Consent Statement:** Informed consent was obtained from all subjects involved in the study.

**Data Availability Statement:** Not applicable.

**Acknowledgments:** The authors would like to thank Sühan Reva for his assistance in regard to the SPSS data analysis.

**Conflicts of Interest:** The authors declare no conflict of interest.

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
