# Peer review of "Road Rage as a Type of Violation of Well-Being in Traffic: The Case of Turkey"

_sustainability, doi:10.3390/su15065147_

Round 1

Reviewer 1 Report

The article has several formal and logical shortcomings that significantly lower the quality of the entire research.

In the Introduction section

- lines 27 to 28 - I consider that belief to be almost identical to the first one, and I see no reason to repeat the same statement,

- in my opinion, the entire paragraph from lines 25 to 30 is not important and does not introduce the issue; I would recommend reformulating the beginning,

- line 45 - I don't think entirely that manifestations of traffic violence also include sexual violence,

- lines 45 to 47 - repeating from and in general throughout the entire paragraph and article, I recommend more scientific language,

- line 63 - I don't consider it important to include the information about the baseball bat,

- line 67 - the sentence begins with "these situations", but it is not described what situations the authors specifically mean,

- I would definitely appreciate it if the paragraph from line 76 was more developed; this could be a good introduction because this is where I perceive the beginning of the article,

- the authors do not define acronyms at all, for example, DAS and DAX, and do not specify the approaches in more detail, which ultimately loses its meaning,

- I would probably also reevaluate the placement of the information from line 87 because I think it belongs more to the methodology,

In the Materials and Methods chapter

- I would specify more how the online survey was carried out, where and why, and I would also state the overall motivation for this article, the motivation for choosing the questionnaire, how the respondents were approached, etc.

- line 126 - the authors start sentences with the word "nearly half" and then state 51.1%. Is that not nearly half or more than half, or am I wrong?

- table 1 - the authors could further develop other professions, cities and other data... and not only mention "other",

- lines 142 to 170 - I would include all this in the theoretical background because the authors describe what has already been examined, and I do not consider that to be a methodology,

Results

- overall, I evaluate the results positively from the point of view of the research, the authors used appropriate methods (even if they did not mention them at all in the methodology), but the presentation of the results is wrong,

- the results are presented very chaotically, I personally got lost a couple of times while reading this article, and I read it several times,

- the authors could consider using clear tables or graphs, which would undoubtedly help the article,

- again, the authors do not define abbreviations, so it is impossible to find out what the authors meant,

- maybe it's just me because I prefer tables, pictures, and diagrams... but I think it would be a better presentation,

- I also recommend using a different format of numbers.. instead .74 I recommend 0.74.. ?

Discussion

- in the whole part, I again lack a better presentation,

Conclusion

It needs to answer why this article was important, what it brought, the reference for further research, and what we should learn.

The authors put a lot of work into the current version, but I recommend working on the article a little more.

Author Response

Dear Referee, 

Thank you for your comments and reviews.

Please find attached our explanations/answers on your comments.

Kind Regards,  

Reviewer 2 Report

This method is not particularly new, and the results of the DAS in the Turkish population presented in this paper are easily imaginable. Thus, the information presented in this paper does not appear to be new. However, since road rage is a fairly important issue, the results presented in this paper should be important for future road rage studies. However, there are a few minor points in this paper that should be corrected.

1. There are some typos and garbled texts, therefore the authors need to check the paper again and correct them.

2. The conclusion section seems a bit long, so that part should be shorter and easier to understand.

Author Response

Dear Referee, 

Thank you for your comments and reviews. 

Please find attached our explanation on your comments. 

Kind Regards,   

Reviewer 3 Report

Your paper is a case study on the implementation of an adjusted methodology to assess road rage features and interrelate with socio-demographic variables in a sample of drivers in Turkey. As such, it is not clear how the research contributes to analyze road rage as a type of violation of well-being. It mostly deals with road rage as an observed phenomenon not as a negative impact. Thus, your title is not completely accurate. But I do not propose a change, unless you can come up with one that better serves the purpose of the specific research.

In the title, the phrase: "Turkey example" is not correct and should be replaced with "A case study in Turkey" or something similar.

There are some syntax errors in various parts of your text. Some examples: "Research shows that those high driving anger levels would be expected to become angry more frequently. These situations arouse anger"; "the stress and rage that traffic is causing has too many negative effects on the lives of the people."-here the word "too" should be erased; "A 14-item short form of DAS which was correlated with DAS subscales."; "In Turkey by Esiyok, Yasak, 97 Korkusuz et al. [14], psychometric analysis of the scale..."; "In addition, based on the study in the USA by Shinar & Compton et al. [16], and the study in 102 Malaysia by Kamarudin et al. [17], showed that..." etc. Please read though the text and you may find such errors and fix them. 

In Table 1, "Basic Personal Car" is a confusing term. You might want to erase the word "basic" and check if "privately owned car", "private car" or "personal vehicle" work better.

In 3.5 significant variables are presented in a cumulative, somewhat "difficult" way for the reader. It could be useful to present the results separately for each of the main variables (e.g. age etc.). Furthermore, in addition to Tables, some graphical representations (e.g. charts) could help the reader understand the differences and comparisons between the variables and their effect on the results.

A minor remark is that it is common not to add new references in the conclusive part of the paper, and I suggest to integrate them into the introduction and main part of the paper, while provide shorter and denser conclusions regarding the general aspects and considerations about road rage. Furthermore, you should focus your conclusions more on the methodology that you have used and how feasible and useful its implementation was or/and whether it can support countries/regions/cities to assess and compare driving anger and develop/exchange appropriate counter policies and measures.  

In the reference list, I believe reference 38 is not complete.

Author Response

(The authors gave the same response as above.)

Round 2

Reviewer 1 Report

The article has been edited, and I see an improvement.

Author Response

Dear Reviewer, 

Thank you for your comment. 

I will improve it and resubmit the revised manuscript.

Kind Regards,